# Particulate Organic Carbon in the Tropical Usumacinta River, Southeast Mexico: Concentration, Flux, and Sources

**Daniel Cuevas-Lara [1], Javier Alcocer [2,\*], Daniela Cortés-Guzmán [1], Ismael F. Soria-Reinoso [1], Felipe García-Oliva [3], Salvador Sánchez-Carrillo [4] and Luis A. Oseguera [2]**

[1] Programa de Posgrado en Ciencias del Mar y Limnología, Universidad Nacional Autónoma de México, Av. Universidad 3000, Alcaldía Coyoacán, C. P., Mexico City 04510, Mexico; daniel.cuevas@comunidad.unam.mx (D.C.-L.); dacortesgu@unal.edu.co (D.C.-G.); ismafs21@gmail.com (I.F.S.-R.)

[2] Grupo de Investigación en Limnología Tropical, FES Iztacala, Universidad Nacional Autónoma de México, Av. de los Barrios No.1, Los Reyes Iztacala, Tlalnepantla 54090, Mexico; loseguera@unam.mx

[3] Instituto de Investigaciones en Ecosistemas y Sustentabilidad, Universidad Nacional Autónoma de México, AP 27-3, Santa María de Guido, Morelia, Michoacán 58090, Mexico; fgarcia@cieco.unam.mx

[4] Departamento de Biogeoquímica y Ecología Microbiana, Museo Nacional de Ciencias Naturales (CSIC), Serrano 115 dpdo, 28006 Madrid, Spain; sanchez.carrillo@mncn.csic.es

\* Correspondence: jalcocer@unam.mx; Tel.: +52-55-5623-1333 (ext. 39719)

**Abstract:** Particulate organic carbon (POC) derived from inland water plays an important role in the global carbon (C) cycle; however, the POC dynamic in tropical rivers is poorly known. We assessed the POC concentration, flux, and sources in the Usumacinta, the largest tropical river in North America, to determine the controls on POC export to the Gulf of Mexico. We examined the Mexican middle and lower Usumacinta Basin during the 2017 dry (DS) and rainy (RS) seasons. The POC concentration ranged from 0.48 to 4.7 mg L$^{-1}$ and was higher in the RS, though only in the middle basin, while remaining similar in both seasons in the lower basin. The POC was predominantly allochthonous (54.7 to 99.6%). However, autochthonous POC (phytoplankton) increased in the DS (from 5.1 to 17.7%) in both basins. The POC mass inflow–outflow balance suggested that floodplains supply (C source) autochthonous POC during the DS while retaining (C sink) allochthonous POC in the RS. Ranging between 109.1 (DS) and 926.1 t POC d$^{-1}$ (RS), the Usumacinta River POC export to the Gulf of Mexico was similar to that of other tropical rivers with a comparable water discharge. The extensive floodplains and the "Pantanos de Centla" wetlands in the lowlands largely influenced the POC dynamics and export to the southern Gulf of Mexico.

**Keywords:** POC; TSS; chlorophyll-a; tropical river; Selva Lacandona; Pantanos de Centla; Chiapas; Mexico

## 1. Introduction

During the last two decades, several studies mentioned that inland aquatic ecosystems are important agents in the coupling of biogeochemical cycles between continents, the atmosphere, and oceans (e.g., [1,2]). Rivers play a mediatory function in carbon (C) storage, processing, and release among these compartments. Rivers also have effects on the regional C cycle since they couple landscapes through the lateral export from terrestrial ecosystems [3]. However, integral approaches to the carbon biogeochemistry associations between habitats remain poorly explored in inland water research [2].

Rivers annually deliver approximately 0.5 Pg of organic carbon (OC) to the world's oceans, with nearly half arriving in particulate form [4–6]. The difference between rock-derived and biosphere-derived particulate organic carbon (POC) loads arriving into oceans from terrestrial systems through rivers contributes to the regulation of atmospheric and terrestrial C reservoirs over a geological timescale by sequestration in marine sediments [5,7–9]. The activity of primary producers in rivers may regulate the C inventory

on a shorter scale. For example, the fluvial C inventory may be affected by algal in situ production, increasing the POC (autochthonous) and reducing the $CO_2$ concentrations [10]. Therefore, a considerable fraction of the C dynamics in river ecosystems depends on the variability of POC within the C pool [11], including the temporal and spatial changes occurring in their sources (allochthonous vs. autochthonous).

Seasonal hydrologic variation often guides POC supply and flux in tropical rivers. While low-water discharge conditions enhance autochthonous production and decrease POC flux [12], high-water discharge conditions increase POC flux with the allochthonous terrestrial matter via erosion [13,14]. However, the seasonal effect on the flux and source of POC is not always clear in tropical rivers, as in some mountainous watersheds [15].

Studies on POC seasonal variations usually circumscribe to limited zones of the rivers. This approach prevents a comprehensive understanding of POC dynamics along the fluvial continuum, from headwaters downstream to the river's mouth. For example, the presence of extensive floodplains and wetlands may diminish the POC through transformation to dissolved forms (mineralized as $CO_2$) and may increase it via primary production activity. This affects the incoming and outgoing POC flux [16–18].

Studies regarding riverine POC flux in the tropics—including Latin America—are still limited [19]. Although available data suggest that tropical rivers are important for the transport of C to oceans, POC flux estimates are still uncertain due to the lack of data, the poor representation of spatial and temporal C dynamics, and indirect estimates, such as for suspended matter flux [8]. Therefore, providing information on the spatial variation in the amount and sources of fluvial POC in the tropics in a seasonal framework will help to unravel the functioning of tropical rivers in the regional and global C cycle.

The Usumacinta River is a transboundary river (Mexico–Guatemala) that is among the largest (i.e., in terms of water discharge, length, drainage basin) tropical rivers in North America and the longest in Mesoamerica [20]. Through its river basin, marked biophysical and hydrological shifts occur [21–23]: the Mexican portion of the middle basin is dominated by steep slopes and covered by well-preserved tropical rainforests, such as the Selva Lacandona [24], while a large expanse of floodplain wetlands and shallow lakes (Pantanos de Centla Biosphere Reserve) extend across the lower basin [25]. The strong extreme climate seasonality and complex mosaic of landscapes should exert strong control over both the river hydrology and POC dynamics in the tropical Central America region, but no information on this topic is currently available.

In this study, our primary goal was to determine the effect of hydrologic tropical seasons (rainy versus dry season) on fluvial POC throughout the basin, assessing its transport, processing, and export. To understand this effect, herein we addressed two main questions: (1) Do the POC sources (autochthonous and allochthonous), concentration, and flux change seasonally along the Usumacinta River and its main tributaries? (2) What effect do the floodplain and wetlands located in the lower basin of the Usumacinta River have on the seasonal POC export? We expected (i) a higher proportion of allochthonous POC in the high-relief, forested landscapes of the middle basin compared to the flat, lower basin where better conditions for autochthonous production may exist (lower water current and low turbidity); (ii) an increase in the POC concentration and flux along the river in the rainy season this is associated with a higher erosion and dragging of terrestrial matter; (iii) that floodplains and wetlands add POC to the fluvial export toward the Gulf of Mexico due to their potential algal development in the dry seasons while retaining POC during the rainy season.

Changes in POC concentration, water quality variables, and C/N ratios of particulate matter along with hydrological variables were analyzed throughout two consecutive seasons to unravel the mechanisms that were most likely to be controlling the POC flux and sources in the Usumacinta River basin. This study contributes to improving the understanding of tropical rivers as biogeochemical conduits of continental carbon. Furthermore, this contribution provides the first estimate of POC export from a Mexican river to the southern Gulf of Mexico.

## 2. Materials and Methods

### 2.1. Study Site

The Usumacinta River is located within the Grijalva–Usumacinta Hydrological Basin to the southeast of Mexico and north of Guatemala. The basin lies between 14°55′ and 18°35′ north and 91°20′ and 94°15′ west [26]. The Usumacinta basin covers ≈77,000 km$^2$ of the Guatemalan and Mexican territories [27], while the Grijalva basin covers ≈57,000 km$^2$ [28]. The maximum altitude in the basin is 3800 m [20].

The Usumacinta River originates in the Cuchumatanes mountain range (upper basin), Guatemala, and runs through the Mexican States of Chiapas and Tabasco (middle and lower basins). The adjacent Grijalva River, which is artificially connected to the Usumacinta River before the river mouth at the southern Gulf of Mexico [29], increases the total water discharge through the Usumacinta River's mouth. The maximum order (Strahler method) of both rivers at their confluence is 7 [30]. The mean annual water discharge of both rivers is $100 \times 10^9$ m$^3$, where nearly 64% of this corresponds to the Usumacinta River discharge [31].

The climate is tropical and wet, with a mean annual temperature of between 8 and 12 °C in the highlands (>1000 m altitude) and 26 and 36 °C in the lowlands (<1000 m) [32]. The rainy season (RS) takes place between May and October and the dry season (DS) between November and April [33,34]; the average precipitation for the seasons is 4500 and 200 mm, respectively [35]. During the RS, characteristic tropical storms increase the Usumacinta River's water discharge, causing flooding in the lowlands. The geomorphological units that characterize the Usumacinta basin are the mountainous area in the highlands and the lowland coastal plain [20]. The mountainous area is composed of the Sierra de Los Cuchumatanes (limestone and dolomites from the Cretaceous), the Sierra Madre de Chiapas (diorites and granites from the Paleozoic), and the Altos de Chiapas (marine and continental carbonates from the Mesozoic with volcanic deposits from the Cenozoic). The coastal plain is a low-relief area formed by sedimentary Cenozoic rocks of alluvial origin [36].

Evergreens, coniferous, and oak forests covered 85% of the middle basin in the 2000s, but they have seen a near 14% reduction in area in the last 15 years [37,38]. The replacement of the natural vegetal cover with agricultural land, space for raising cattle, and urban development in the middle basin has caused landslides and floods during extreme rain events [39]. Mangrove forests, along with other hydrophytes, extend across the lower portion of the basin, covering 8% of the Usumacinta–Grijalva basin [40]. Submerged and floating macrophytes are also present in the lakes and floodplains connected to the Usumacinta's main channel [41].

### 2.2. Sampling

Two sampling campaigns were carried out in the Mexican portion of the Usumacinta River (middle and lower basins); the first was conducted at the end of the DS in May 2017 and the second was conducted in the RS in November 2017, just after the rainiest month (October). We selected seven sites along the main channel and major tributaries of the Usumacinta River: 4 in the middle (M1 to M4) and 3 in the lower basin (LU, LG, and LPP; Figure 1).

M1 corresponds to the Lacantún River, which drains 15,772 km$^2$ of the highlands of Chiapas. From this site, the course stretches 656 km to the river mouth. M2, the Tzendales River, was chosen as a reference site due to its undisturbed and pristine condition, as its water emerges and crosses through the protected area of the Lacandona rainforest [37]. M2's watershed covers 1487 km$^2$, and from this site, the river stretches 653 km to the river mouth. Both M1 and M2 are in the Montes Azules Biosphere Reserve tropical rainforest. M3 (Frontera Corozal) and M4 (Boca del Cerro) correspond to sites along the main channel of the Usumacinta River in the middle basin (Figure 1). They have watersheds of 46,564 and 51,064 km$^2$ and lengths of 525 and 385 km to the river mouth, respectively. LU (Rivera Alta) is the main channel of the Usumacinta River in the lower basin. It stretches 25 km to the river mouth, while its watershed spans 70,350 km$^2$. LG refers to the Grijalva River



before its confluence with the Usumacinta River. We used it as a reference to the adjacent Grijalva watershed, which drains 57,272 km$^2$ of land. This site is also 25 km away from the river mouth. LPP corresponds to the distributary San Pedro–San Pablo River; it is 7 km away from the river mouth and drains 72,569 km$^2$. LPP represents the second and smaller river mouth of the Usumacinta River. The three lower basin sites are in the Ramsar wetland and Biosphere Reserve "Pantanos de Centla," which is the largest (3027 km$^2$) wetland area of North America and the third-largest in Latin America ([42]; Figure 1).

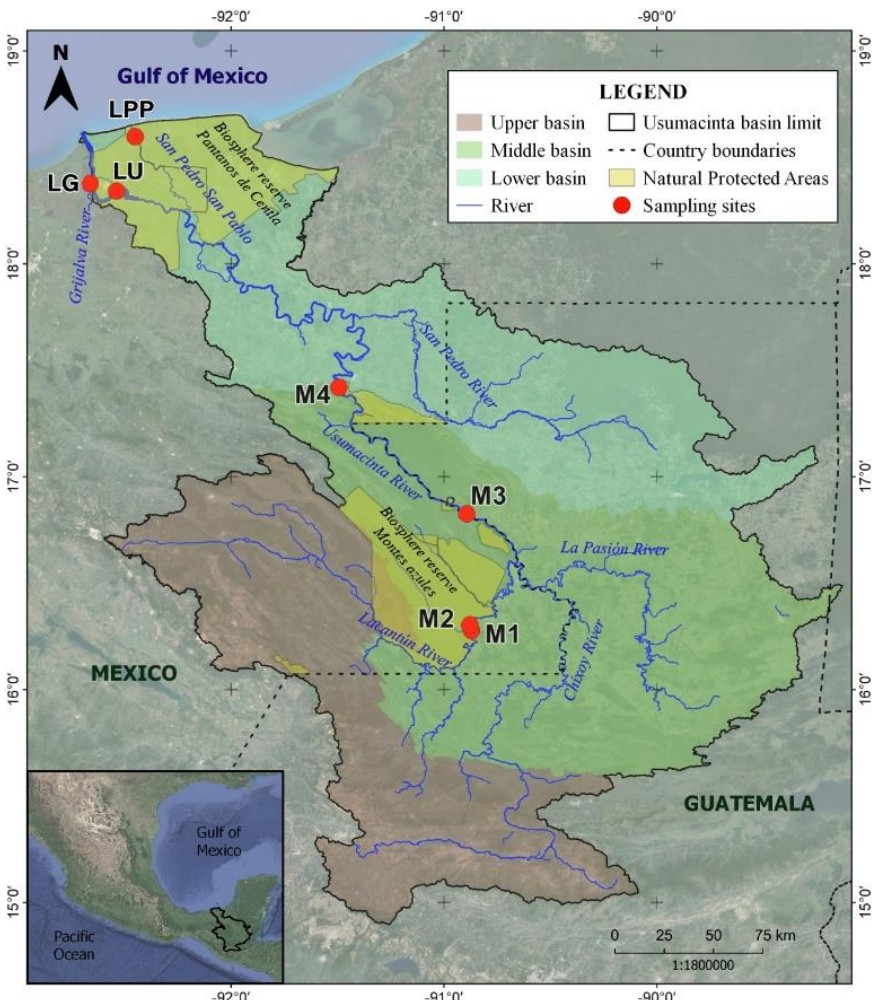

**Figure 1.** The Usumacinta River basin and sampling sites (elaborated using the INEGI database; datum: WGS84).

At each sampling site, a cross-section profile was recorded with a Garmin GPSMap 526S Echo Sounder. Vertical profiles (every meter) of temperature (T), dissolved oxygen concentration (DO), pH, electrical conductivity ($K_{25}$), turbidity (Turb), and oxidation/reduction potential (ORP) were determined with a Hydrolab DS5 multiparameter sonde (OTT Hydromet, Kempten, Germany). The mean current velocity was measured with a Swoffer 3000 current meter (Swoffer, Sumner, Washington, USA). Three sub-superficial (1/3 maximum depth) water samples were collected along the cross-section of the river (center and 1/3 toward both sides across the river section) in all sites but M2; based on the M2 reduced width (40 m), only one water sample was taken at the center. The water discharge (Q, m$^3$ s$^{-1}$) was calculated using cross-section profiles and the measured mean river flow velocities and then normalized by the watershed area upstream of the sampling point (L km$^2$ s$^{-1}$). Water samples for the POC, total suspended solids (TSS), and chlorophyll a (Chl-a) concentrations were taken with a horizontal Van-Dorn-type sampling bottle.

### 2.3. Analytical Methods

The Chl-a concentration measurement followed the EPA 445.0 method [43]. Triplicate water samples were filtered through Whatman GF/F glass microfiber filters (nominal 0.7 μm pore size). A total of 10 mL of 90% acetone was added to each filter. Chl-a was extracted at 4 °C in dark conditions for 12–24 h. After centrifugation (2700 rpm/15 min), the Chl-a extracted from the supernatant was measured with a digital fluorometer TD 10-AU (Turner Designs, San Jose, CA, USA, country). To measure the TSS concentration, duplicate water samples (300 to 1000 mL) were pre-filtered through 100 μm pore mesh to remove "large swimmers" and were furthered retained in pre-combusted (550 °C/4 h) and pre-weighed Whatman GF/F filters. The filters were then dried for at least 48 h (50 °C) and weighed again.

Duplicate water samples (20 to 80 mL) were filtered through pre-combusted (550 °C/4 h) Whatman GF/F filters. Filters were further acidified (HCl 10%) to eliminate inorganic C [44], folded in clean pre-baked aluminum foil, and frozen until analysis. The C and N content in the filters was measured via elemental analysis (Elemental Analyzer Carlo Erba NC2100, CE Elantech, Lakewood, NJ, USA) to calculate the POC concentration and the particulate carbon-to-nitrogen ratio (POC/PN). The POC concentrations and Q data were used to calculate instant POC fluxes (FPOC, t C d$^{-1}$): Q × POC concentration. We also calculated the POC yields using (YPOC, kg C km$^2$ d$^{-1}$) Q × POC concentration/watershed area.

### 2.4. Data Analysis: POC Sources, Mass Balance, and Statistical and Geographical Analyses

Chl-a, TSS, and POC/PN were used to provide insight into the POC sources (autochthonous vs. allochthonous). The contribution of POC to the total particulate matter load (POC/TSS) was used to determine the allochthonous origin of POC (i.e., rock or soil; [9,45]); we calculated the POC contribution as a percentage. We used the Chl-a concentration to obtain the algal carbon from phytoplankton ($C_{ALG}$). The $C_{ALG}$ was estimated by considering that 1 mg of Chl-a = 40 mg of $C_{ALG}$ [14,46]. To obtain the autochthonous proportion, the $C_{ALG}$ to POC ($C_{ALG}$ /POC) and TSS ($C_{ALG}$ /TSS) proportions were calculated as percentages. The latter was used to determine the seasonal change in POC sources, along with the POC/TSS. The POC/PN ratio was used to provide insight into the relative signals of decomposition or type of particulate matter [47].

The mass inflow–outflow balance of POC in the lower basin was calculated according to [48] using the difference between the M4 site (input), which is the last site of the middle basin, and the two Usumacinta River mouths (LU and LPP, output) at the lower basin. Thus, the difference in POC fluxes between the outputs and inputs determines whether the zone stores (sink, positive values) or supplies (source, negative values) POC, but without any implication regarding the process or fate of the C [19].

The seasonal differences in POC concentrations, POC flux, and POC/PN were evaluated using *t*-tests and non-parametric Mann–Whitney U-tests; U-tests were performed using groups with the same frequency distribution shape. The spatial variability of water quality was examined using principal component analysis (PCA). The PCA was performed using a correlation matrix. The variables were standardized to find the unit sample variance, which is the sample mean subtracted from values and divided by the standard deviation. We computed the distance to the river mouth with a map projection using the WGS84 datum. The limits of the catchments and lengths of the sampling sites were determined with digital elevation models from the USGS dataset [49] using the software QGIS Geographic Information System [50] with GRASS (v. 7.8.5). Non-parametric Spearman rank-order correlation (ρ) and linear regressions were performed to detect associations between the POC and water quality variables. All the statistical procedures were carried out in Sigmaplot version 14.0. An alpha level of 0.05 was used for all statistical tests.

## 3. Results

### 3.1. River Water Quality Dynamics

The Usumacinta River had significant seasonal changes in Q and maximum depth (Zmax), with the highest values recorded in the RS. The Q ranged between 1 and 20 L km$^{-2}$ s$^{-1}$ (10 ± 6 L km$^{-2}$ s$^{-1}$, average ± SD) in the DS and varied from 5 to 123 L km$^{-2}$ s$^{-1}$ (72 ± 46 L km$^{-2}$ s$^{-1}$) in the RS (Table 1). M3 and M4 had the most significant seasonal changes in Q and Zmax (Table 1). The middle basin sites had a higher Q than the lower basin sites due to the watershed. In the DS, the Lacandona rainforest sites had the highest Q (M1 and M2; >10 L km$^{-2}$ s$^{-1}$), while M2, M3, and M4 had the highest Q in the RS (>100 L km$^{-2}$ s$^{-1}$). The LPP site had the lowest Q in both seasons.

**Table 1.** Average (first line) and standard deviation (second line) values of water quality variables in the Usumacinta River main channel and major tributaries in the dry and rainy seasons.

| Season | River/Site | | Q | Zmax | T | DO | pH | K$_{25}$ | ORP | Turb | Chl-a | TSS |
|---|---|---|---|---|---|---|---|---|---|---|---|---|
| | Lacantún/M1 + | X$^-$ | 312 | 3.5 | 27.3 | 6.3 | 6.9 * | 395 | 208 | 46 | 4.7 | 49 |
| | | σ | | 0 | <0.1 | 1.1 | 2 | 2 | 8 | 0.5 | 7 |
| | Tzendales/M2 + | X$^-$ | 23 | 1.5 | 27.8 | 7.2 | 7.1 * | 931 | 202 | 10 | 0.9 | 14 |
| | | σ | | 0 | 0 | 1.0 | 0 | <1 | 1 | 0.1 | 1 |
| | Usumacinta/M3 | X$^-$ | 396 | 4.2 | 30.7 | 6.5 | 7.1 * | 706 | 197 | 11 | 3.0 | 12 |
| | | σ | | 0.1 | <0.1 | 1.1 | <1 | 8 | 1 | 0.4 | 1 |
| | Usumacinta/M4 | X$^-$ | 369 | 20.1 | 30.7 | 6.4 | 7.2 * | 839 | 169 | 7 | 1.9 | 7 |
| Dry | | σ | | 0.1 | <0.1 | 1.2 | 3 | 3 | 1 | 0.4 | 1 |
| Season | Usumacinta/LU | X$^-$ | 432 | 12.5 | 29.9 | 3.3 | 7.4 * | 33,368 | 153 | 8 | 8.3 | 12 |
| | | σ | | 0.9 | 2.1 | 1.1 | 18,806 | 14 | 3 | 0.8 | 2 |
| | Grijalva/LG + | X$^-$ | 527 | 11 | 30.3 | 4.0 | 7.4 * | 27,610 | 142 | 6 | 4.0 | 40 |
| | | σ | | 0.8 | 2.2 | 1.2 | 15,126 | 5 | 1 | 1.0 | 15 |
| | San Pedro–San Pablo/LPP – | X$^-$ | 78 | 4.8 | 29.8 | 5.1 | 7.6 * | 57,098 | 142 | 132 | 17.6 | 54 |
| | | σ | | | 0.3 | 0.8 | 1.1 | 1148 | 5 | 85 | 2.2 | 11 |
| | Total | X$^-$ | 305 | 8.2 | 29.6 | 5.5 | 7.2 * | 13,019 | 174 | 23 | 6.3 | 28 |
| | | σ | 187 | 6.6 | 2.2 | 1.9 | 1.6 | 19,866 | 26 | 36 | 5.5 | 20 |
| | Lacantún/M1 + | X$^-$ | 948 | 6.3 | 22.2 | 8.5 | 8.1 * | 374 | 346 | 100 | 0.4 | 119 |
| | | σ | | <0.1 | <0.1 | 1.1 | <1 | 16 | 4 | 0.1 | 14 |
| | Tzendales/M2 + | X$^-$ | 173 | 3.8 | 23.4 | 8.0 | 8.1 * | 539 | 355 | 8 | 0.1 | 16 |
| | | σ | | <0.1 | <0.1 | 1.0 | <1 | 1 | 1 | 0.0 | |
| | Usumacinta/M3 | X$^-$ | 5715 | 15.6 | 24.3 | 5.8 | 7.7 * | 316 | 357 | 55 | 0.8 | 76 |
| | | σ | | <0.1 | 0.1 | 1.1 | 5 | 9 | 1 | 0.1 | 6 |
| | Usumacinta/M4 | X$^-$ | 5934 | 35.2 | 24.9 | 6.6 | 7.8 * | 359 | 428 | 52 | 0.7 | 71 |
| Rainy | | σ | | <0.1 | <0.1 | 1.0 | <1 | 4 | 2 | 0.1 | 2 |
| Season | Usumacinta/L1 | X$^-$ | 2709 | 17.1 | 26.7 | 4.6 | 7.7 * | 361 | 409 | 44 | 3.4 | 63 |
| | | σ | | <0.1 | 0.1 | 1.0 | <1 | 2 | 8 | 0.3 | 7 |
| | Grijalva/LG + | X$^-$ | 2618 | 12.3 | 27.9 | 1.4 | 7.3 * | 337 | 381 | 49 | 3.3 | 82 |
| | | σ | | <0.1 | 0.1 | 1.1 | 1 | 10 | 9 | 0.3 | 22 |
| | San Pedro–San Pablo/LPP – | X$^-$ | 368 | 4.8 | 27.5 | 1.9 | 7.4 * | 394 | 390 | 43 | 3.7 | 69 |
| | | σ | | | 0.1 | 0.2 | 1.1 | 14 | 4 | 15 | 0.5 | 20 |
| | Total | X$^-$ | 2638 | 14 | 25.4 | 5.2 | 7.7 * | 360 | 390 | 55 | 2.0 | 78 |
| | | σ | 2395 | 11 | 1.9 | 2.3 | 1.7 | 37 | 31 | 21 | 1.5 | 25 |

Q: water discharge (m$^3$ s$^{-1}$); Zmax: maximum depth (m); T: temperature (°C); DO: dissolved oxygen concentration (mg L$^{-1}$); K$_{25}$: electrical conductivity standardized at 25 °C (μS cm$^{-1}$); ORP: oxidoreduction potential (mV); Turb: turbidity (NTU); Chl-a: chlorophyll-a concentration (μg L$^{-1}$); TSS: total suspended solids (mg L$^{-1}$); * geometric mean. + tributary; – distributary.

The total Usumacinta River Q into the Gulf of Mexico considering both Usumacinta River mouths (LU and LPP) and the Grijalva River (LG) was 1037 m$^3$ s$^{-1}$ in the DS and 5695 m$^3$ s$^{-1}$ in the RS. The Usumacinta River (LU plus LPP) contributed 49% and 54% to the total Q in the DS and RS, while the Grijalva contributed 51% and 46%, respectively. However, the Grijalva had a slightly higher watershed-normalized Q than the Usumacinta in both seasons, with 9 L km$^{-2}$ s$^{-1}$ in the DS and 46 L km$^{-2}$ s$^{-1}$ in the RS against 7 and 44 L km$^{-2}$ s$^{-1}$, respectively.

The water temperature at all the sites was significantly higher in the DS than in the RS (*t*-test; $p < 0.001$). The DO was similar in both seasons; it ranged from 3.3 to 7.2 mg $L^{-1}$ in the DS ($5.5 \pm 1.7$ mg $L^{-1}$) and from 1.4 to 8.5 mg $L^{-1}$ in the RS ($5.2 \pm 2.3$ mg $L^{-1}$). The DO decreased from the uppermost site toward the river mouth in both seasons; the middle basin sites averaged above 6 mg $L^{-1}$, while the lower basin averaged concentrations below 5 mg $L^{-1}$. The pH in the Usumacinta River ranged from 6.9 to 8.1; it rose from 6.9 (middle) to 7.3 (low basin) in the DS and decreased from 8.1 (middle) to 7.5 (low basin) in the RS (Table 1).

Seawater intrusion influenced the $K_{25}$ values in the DS due to a conspicuous saltwater wedge detected in the lower basin sites' vertical profiles. The $K_{25}$ in the RS was lower than in the DS. In the middle basin sites, $K_{25}$ did not exceed 1000 μS $cm^{-1}$ in both seasons (Table 1). Overall, the Usumacinta River behaved as an oxidizing environment. Seasonally, the river had lower ORP values in the DS ($174 \pm 26$ mV (142 to 208 mV)) than in the RS ($390 \pm 31$ mV (355 to 428 mV)). Turb ranged from 6 to 132 NTU ($21 \pm 29$ NTU) in the DS. In this season, LPP noticeably displayed high values and variation (Table 1). In the RS, Turb ranged from 8 to 100 NTU ($55 \pm 21$ NTU). The average turbidity was higher in the RS in all river sites, except in M2 and LPP (Table 1).

The Chl-a concentration along the Usumacinta River varied from 0.1 to 17.6 μg $L^{-1}$ ($4.2 \pm 4.6$ μg $L^{-1}$). In general, the Chl-a was lower in the RS than in the DS; in the DS, the Chl-a concentration ranged from 0.8 to 20.2 μg $L^{-1}$ ($6.3 \pm 5.5$ μg $L^{-1}$), while it ranged from 0.1 to 4.4 μg $L^{-1}$ in the RS ($2.0 \pm 1.5$ μg $L^{-1}$). The TSS concentration varied seasonally in the Usumacinta River, and higher values were recorded in the RS. The mean TSS concentration in the RS ($77 \pm 27$ μg $L^{-1}$) nearly doubled in the DS ($28 \pm 21$ μg $L^{-1}$; Table 1).

As shown by the PCA (Figure 2), three principal components explained 84.6% of the river water quality variance. PC 1 correlated inversely with temperature and directly with ORP and TSS. PC 2 directly correlated with the Turb, Chl-a, and $K_{25}$. PC 3 was inversely correlated with DO.

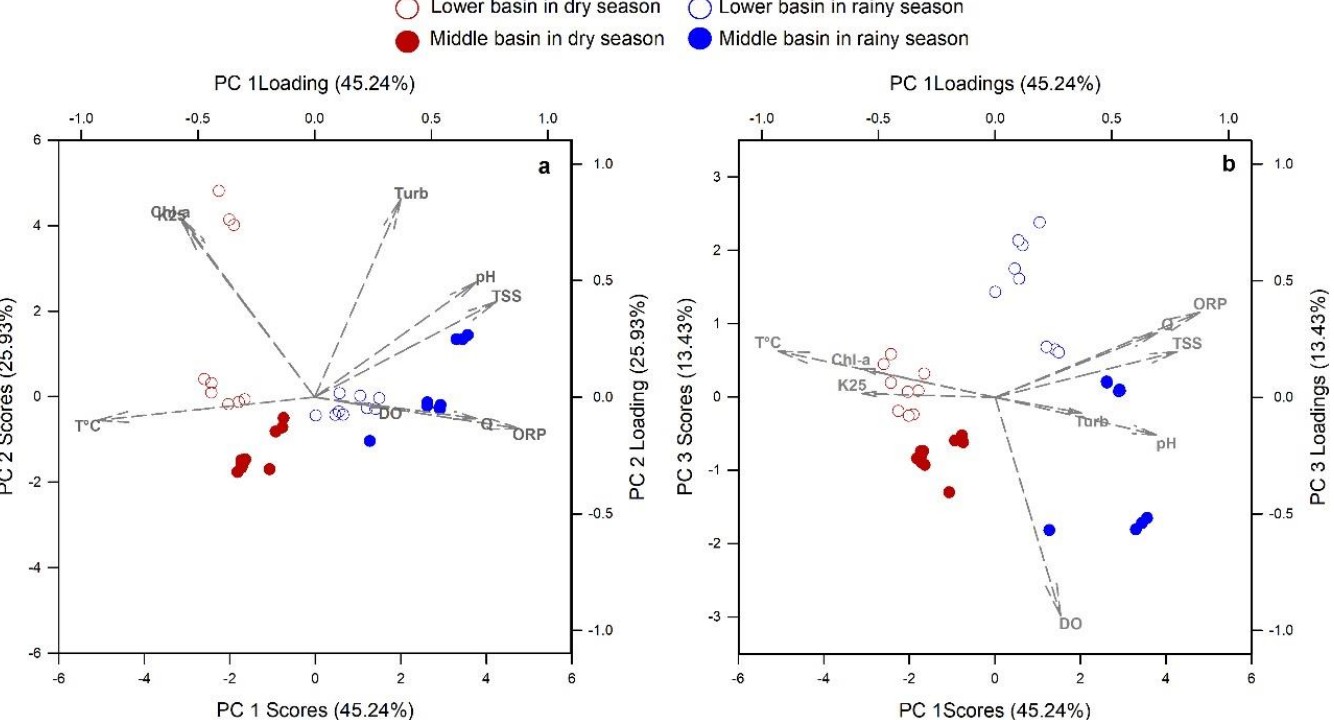

**Figure 2.** Spatial and seasonal water quality variation in the Usumacinta River main channel and major tributaries, as displayed using PCA: (**a**) Representation of PC 1 against PC 2 and (**b**) representation of PC 1 against PC 3. The percentages indicate the explained variance of the PCs.

The PC 1 displayed the seasonality of the Usumacinta River. In the DS, the river was less oxidant, warmer, and had lower concentrations of TSS and a lower flow rate than in the RS. The PC 1 arrangement also displayed warmer and more oxidant water, with lower TSS concentrations in the lower basin in the RS, except in M2 (Figure 2a). The PC 2 and PC 3 arrangements separated the observations into middle and lower basins in both seasons. In them, it is shown that the lower basin had a higher Chl-a concentration and salinity and less oxygenated waters than the middle basin in both seasons (Figure 2b). The LPP in the DS had very high Turb, $K_{25}$, and Chl-a concentrations (Table 1).

### 3.2. POC Relationship with Water Quality Variables

The POC concentration was positively correlated with the TSS concentration (linear regression, $p < 0.001$; Figure 3) and positively correlated with Turb ($\rho = 0.39$, $p = 0.016$). In contrast, water discharge was not correlated with POC ($\rho = -0.16$, $p = 0.333$), although TSS was positively correlated with Q ($\rho = 0.47$, $p < 0.05$). In the DS, the POC concentration was positively correlated with the Chl-a concentration ($\rho = 0.83$, $p < 0.001$).

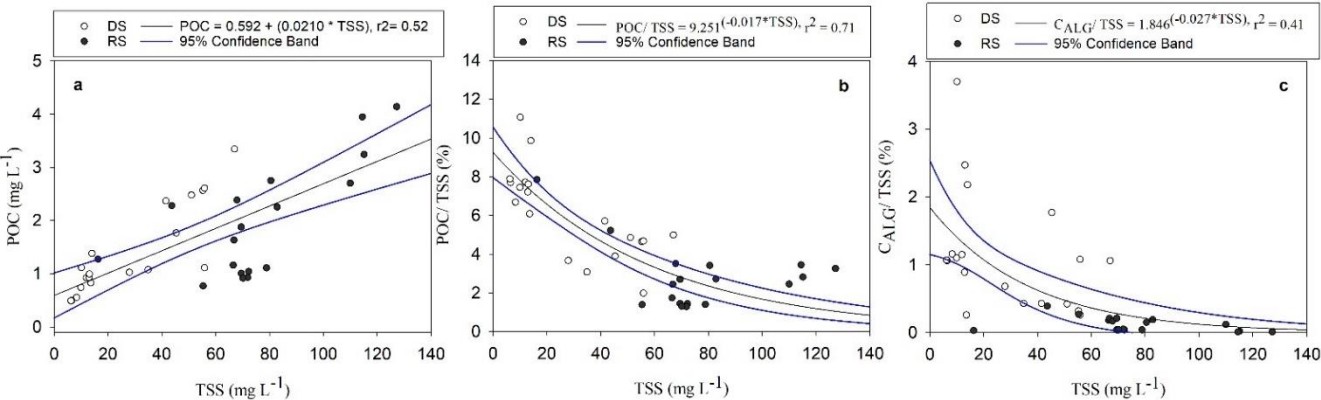

**Figure 3.** Variation in the organic carbon contribution to the total suspended solids in the Usumacinta River during the dry (DS) and rainy (RS) seasons: (**a**) relationship between the POC and TSS concentrations, (**b**) relative content of POC in TSS as a function of the TSS concentration, and (**c**) relative contribution of $C_{ALG}$ to TSS as a function of the TSS concentration.

The percentage of POC in the total particulate constituents (POC/TSS) varied from 1.3 to 11.1%. The DS recorded higher percentages of POC compared with the RS (*t*-test, $p < 0.001$). There was a non-linear relationship between the POC/TSS and the TSS concentration (Figure 3). M2 showed the highest POC/TSS (7.8%) in the RS due to having the lowest TSS concentration (16 mg $L^{-1}$) compared to the rest of the sites (> 60 mg $L^{-1}$; Table 1). The percentage of algal biomass of TSS ($C_{ALG}$/TSS) varied from 0.01 to 3.7%. It also exhibited a lower percentage in the RS, displaying a non-linear relationship with the TSS concentration (Figure 3).

The $C_{ALG}$/POC averaged $11.6 \pm 10.2\%$. It averaged $17.7 \pm 10.3\%$ in the DS and $5.1 \pm 4.6\%$ in the RS. Although the $C_{ALG}$/POC was higher in the DS compared to in the RS (U-test, $p < 0.001$), it remained below 50% (Figure 4) in both seasons. The algal contribution to the POC was higher in the lower basin in both seasons (DS: *t*-test, $p < 0.05$; RS: U-test, $p < 0.001$; Figure 4).

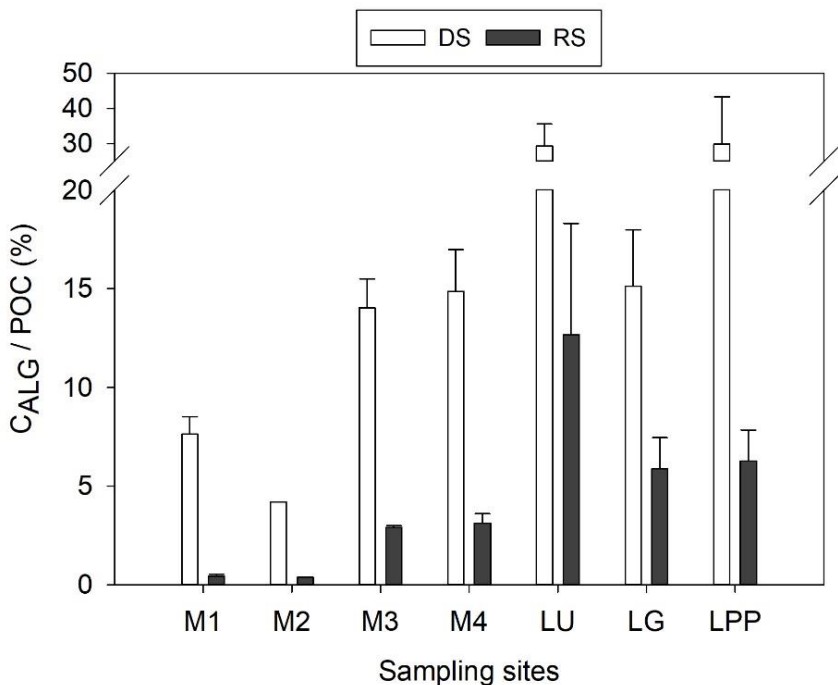

**Figure 4.** Contribution of algal biomass to POC throughout the Usumacinta River basin during the dry (DS) and rainy (RS) seasons. Bars represent averages and the whiskers represent the standard deviations.

The POC/PN ratio averaged $10 \pm 3$ for the DS and $13 \pm 17$ for the RS, but there were no significant differences between the seasons (U-test, $p > 0.05$). The POC/PN ratios in the DS were different in the middle and lower basins (Figure 5).

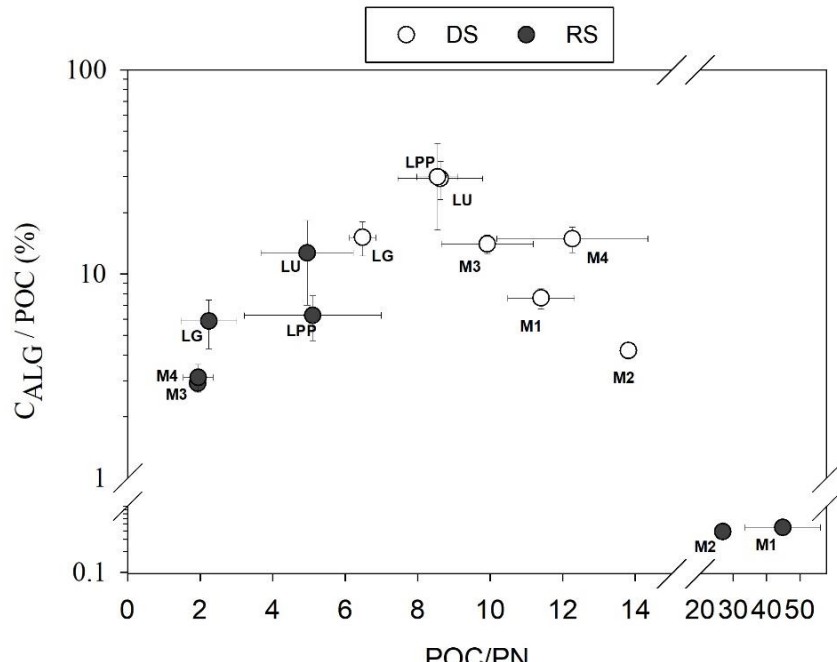

**Figure 5.** Mean algal biomass contribution to POC in the Usumacinta River as a function of the mean POC/PN ratio. The whiskers represent the standard deviations.

### 3.3. Variation of POC Concentration and Yield

The Usumacinta River's POC concentration ranged from 0.48 to 4.7 mg L$^{-1}$ (1.69 $\pm$ 1.03 mg L$^{-1}$). Overall, it was significantly higher in the RS (2.06 $\pm$ 1.14 mg L$^{-1}$) than in the DS (1.42 $\pm$ 0.86 mg L$^{-1}$; U-test, $p > 0.01$). While the POC concentrations in M3, LU, and LPP remained similar in both seasons ($t$-test, $p > 0.05$), the POC concentrations in M1, M2, M4, and LG were between 50 and 80% higher in the RS (Figure 6).

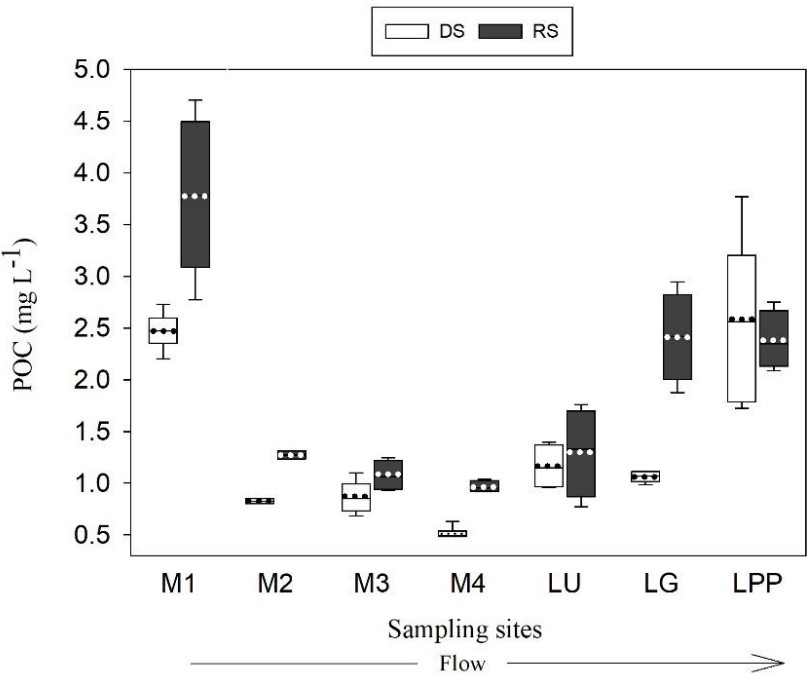

**Figure 6.** Box and whisker plot showing the seasonal and spatial distribution of the POC concentration in the Usumacinta River basin. Boxplots represent the 25th and 75th percentiles of the concentration, and the dotted line within each box represents the mean. The whiskers represent the standard deviation.

In both seasons, the POC followed the same dynamic throughout the river. In the middle basin, the POC diminished downstream from M1 (>2.2 mg L$^{-1}$) to the main channel (0.48 to 1.24 mg L$^{-1}$ in M3 and M4). M2 showed similar lower POC values compared to the main channel (<1.63 mg L$^{-1}$). In the lower basin, the POC increased, especially in the LPP site, which exhibited concentrations above 1.72 mg L$^{-1}$.

The POC yield throughout the Usumacinta River ranged between 0.2 and 19.6 kg km$^2$ d$^{-1}$ (5.5 $\pm$ 6.1 kg km$^2$ d$^{-1}$), with lower values in the DS than in the RS ($t$-test, $p < 0.05$). The POC yield increased from approximately four times to thirty times in the RS (Table 2). In the DS, only the sampling sites located in the Lacandona rainforest (M1 and M2) surpassed 1 kg km$^2$ d$^{-1}$. The yield in the RS always exceeded 1 kg km$^2$ d$^{-1}$ across all sampling sites, but clearly decreased downriver (Figure 7). While in the DS, the Usumacinta River POC yields were similar to the Grijalva River values, in the RS, the high POC concentration of the Grijalva River resulted in a higher POC yield (4.3 $\pm$ 1.2 kg km$^2$ d$^{-1}$ in the Usumacinta versus 9.5 $\pm$ 1.5 kg km$^2$ d$^{-1}$ in the Grijalva).

**Table 2.** Average (first line) and standard deviation (second line) values of POC concentration, POC flux, POC/TSS, POC/PN, and $C_{ALG}$/POC ratios in the sampling sites of the Usumacinta River in the dry and rainy seasons.

| Season | River/Site | | POC Concentration (mg L$^{-1}$) | POC Flux (t C d$^{-1}$) | POC Yield (kg C km$^2$ d$^{-1}$) | POC/TSS (%) | POC/PN | $C_{ALG}$/POC (%) |
|---|---|---|---|---|---|---|---|---|
| Dry Season | Lacantún/M1 + | $X^-$ | 2.47 | 66.7 | 4.2 | 5.1 | 14 | 7.6 |
| | | σ | 0.16 | 4.4 | 0.3 | 0.6 | 2 | 0.9 |
| | Tzendales/M2 + | $X^-$ | 0.83 | 1.7 | 1.1 | 6.1 | 11 | 4.2 |
| | | σ | 0.03 | 0.1 | 0.0 | | 1 | |
| | Usumacinta/M3 | $X^-$ | 0.86 | 29.5 | 0.6 | 7.5 | 10 | 14.0 |
| | | σ | 0.14 | 4.7 | 0.1 | 0.3 | 1 | 1.5 |
| | Usumacinta/M4 | $X^-$ | 0.51 | 16.4 | 0.3 | 7.4 | 12 | 14.9 |
| | | σ | 0.05 | 1.7 | 0.0 | 0.6 | 2 | 2.1 |
| | Usumacinta/LU | $X^-$ | 1.16 | 43.4 | 0.6 | 9.5 | 9 | 29.4 |
| | | σ | 0.18 | 6.8 | 0.1 | 1.8 | 1 | 6.2 |
| | Grijalva/LG + | $X^-$ | 1.06 | 48.4 | 0.8 | 2.9 | 6 | 15.1 |
| | | σ | 0.05 | 2.1 | 0.0 | 0.9 | 1 | 2.9 |
| | San Pedro–San Pablo/LPP – | $X^-$ | 2.57 | 17.3 | 0.2 | 4.5 | 9 | 29.9 |
| | | σ | 0.73 | 4.9 | 0.1 | 0.6 | 0 | 13.4 |
| | Total | $X^-$ | 1.42 | 33.3 | 1.1 | 3.7 | 10 | 16.4 |
| | | σ | 0.86 | 21.1 | 1.4 | 3.2 | 2 | 4.8 |
| Rainy Season | Lacantún/M1 + | $X^-$ | 3.78 | 309.3 | 19.6 | 3.2 | 45 | 0.4 |
| | | σ | 0.66 | 54.3 | 3.4 | 0.3 | 11 | 0.1 |
| | Tzendales/M2 + | $X^-$ | 1.27 | 19 | 12.8 | 7.8 | 27 | 0.4 |
| | | σ | 0.04 | 0.6 | 0.4 | | 1 | |
| | Usumacinta/M3 | $X^-$ | 1.07 | 530.7 | 11.4 | 1.4 | 2 | 2.9 |
| | | σ | 0.13 | 63.6 | 1.4 | 0.0 | 0 | 0.1 |
| | Usumacinta/M4 | $X^-$ | 0.96 | 494.7 | 9.7 | 1.4 | 2 | 3.1 |
| | | σ | 0.05 | 23.9 | 0.5 | 0.1 | 0 | 0.5 |
| | Usumacinta/LU | $X^-$ | 1.30 | 304 | 4.3 | 1.9 | 5 | 12.7 |
| | | σ | 0.37 | 87.1 | 1.2 | 0.5 | 1 | 5.6 |
| | Grijalva/LG + | $X^-$ | 2.42 | 546.3 | 9.5 | 2.9 | 2 | 5.9 |
| | | σ | 0.38 | 85.6 | 1.5 | 0.6 | 1 | 1.6 |
| | San Pedro–San Pablo/LPP – | $X^-$ | 2.38 | 75.9 | 1.0 | 3.8 | 5 | 6.3 |
| | | σ | 0.24 | 7.7 | 0.1 | 1.3 | 2 | 1.6 |
| | Total | $X^-$ | 2.06 | 323.0 | 9.8 | 1.9 | 10 | 4.5 |
| | | σ | 1.14 | 198.9 | 6.0 | 2.1 | 14 | 2.1 |

+ tributary; – distributary.

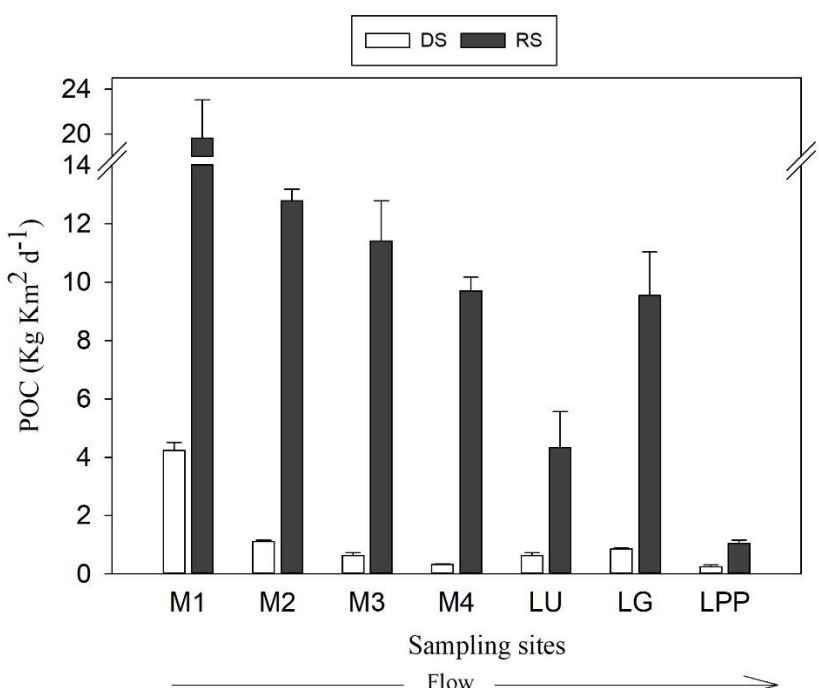

**Figure 7.** Seasonal and spatial distributions of POC yield in the Usumacinta River basin. Bars represent average yields and whiskers represent the standard deviations.

*3.4. Mass Balance of POC in the Usumacinta Lower Basin and POC Export to the Southern Gulf of Mexico*

In the DS, the POC flux increased (3.7 times) from the middle basin (M4) to the lower basin (LU plus LPP). This represented an increase in POC of approximately 70% in the lower Usumacinta River basin. In contrast, the POC mass inflow–outflow balance in the RS was $-114.9$ t C d$^{-1}$. Hence, the lower basin retained near one-fifth of the incoming POC flux from the middle basin (494 t C d$^{-1}$; Table 3).

**Table 3.** Input–output balances of water discharge (Q) and POC in the Usumacinta lower basin and the total POC export to the Gulf of Mexico in the rainy (RS) and dry (DS) seasons of 2017. M4 is the output of the middle basin; LU: Usumacinta River; LG: Grijalva River, LPP: San Pedro–San Pablo River. See Figure 1 for the detailed locations of the sampling sites.

| | Site | Q (m$^3$ s$^{-1}$) | | POC Flux (t C d$^{-1}$) | |
|---|---|---|---|---|---|
| | | **DS** | **RS** | **DS** | **RS** |
| Input | M4 | 369 | 5934 | 16.4 | 494.7 |
| Output | LU | 432 | 2709 | 43.4 | 304.0 |
| | LPP | 78 | 368 | 17.3 | 75.9 |
| Lower Usumacinta basin balance(input-output) | M4 − (LU + LPP) | −141 | 2857 | −44.3 | 114.9 |
| Usumacinta basin export | LU + LPP | 510 | 3077 | 60.7 | 379.9 |
| Usumacinta–Grijalva basin export | LU + LG + LPP | 1037 | 5695 | 109.1 | 926.1 |

In the DS, the POC export to the southern Gulf of Mexico was 109.1 t C d$^{-1}$ and it ranged from 95.7 to 124.5 t C d$^{-1}$. The RS was 926.1 t C d$^{-1}$, which was almost nine times higher than that in the DS (Table 3), and it varied from 676.8 to 1080.4 t C d$^{-1}$. The relative contribution to the total export of each of the rivers located in the lower basin was similar in both seasons. In the RS and DS, the Grijalva River (LG) contributed 44% and 50%, the

Usumacinta River contributed 40% and 33%, and the San Pedro–San Pablo River (LPP) contributed 16% and 8%, respectively.

## 4. Discussion

*4.1. Seasonal Hydrologic Differences in the Usumacinta River Basin and its Implication in POC Sources*

Water discharge in the Usumacinta River revealed a strong seasonal hydrological contrast during rainy and dry periods. The Usumacinta River discharge to the Gulf of Mexico increased by five times in the RS compared to the DS. This increase is noteworthy compared to that of other large rivers, such as the Amazon, Congo, Tana, Yellow, and Mississippi, which exhibited lower seasonal differences from 1.2 to 3.2 times [44,51–54].

The relationship observed between TSS and water discharge clearly indicates seasonal erosion by runoff, as cited in other basins worldwide [6,55]. The cyclonic storms occurring in the Usumacinta–Grijalva basin during the RS promote the episodic movement of large volumes of organic and mineral soil through the watersheds, resulting in hyperconcentrated and highly turbid flows in the main channel of the Usumacinta River (>5% sediment volume [36]).

TSS and turbidity were positively correlated with the POC concentrations, but surprisingly not with Q. Our results showed the positive correlations between two but not all three variables (Q, TSS, and POC) commonly found in fluvial systems, which usually indicate seasonal POC inputs from terrestrial allochthonous sources (e.g., soil erosion; [10,53,56,57]). Therefore, the correlation between POC and Q in the Usumacinta basin was possibly affected by the dilution effect between the river sites. Likewise, the moderate difference between seasons regarding the POC concentration in the lower basin, which was probably associated with the algal activity, also affected the POC and Q relationship.

The non-linear relationship found in the Usumacinta basin between POC/TSS and the TSS's concentration has been observed in other studies worldwide (e.g., [5,58]). This relationship typically results from two different hydrodynamic processes for POC supply: (a) the RS is a period of high soil erosion—allochthonous POC—during flooding events (e.g., topsoil with higher POC/TSS to deep layers with low POC/TSS), (b) the low water level and low turbidity recorded during the DS promote high aquatic primary productivity—autochthonous POC (high POC/TSS) [45]. Most studies in tropical rivers using both basin-scale approaches (e.g., [53,59,60]) and river-site-specific sampling (e.g., [10]) mention the positive effect of algal productivity in the POC supply process.

On the one hand, an intensification of algal development occurred in the DS since the POC was directly related to Chl-a; the $C_{ALG}$/POC increased in all sampling sites and we also found that $C_{ALG}$/TSS explained a significant part of the variation in POC/TSS ($\approx$ 40%, $\rho = 0.625$, $p = 0.004$). Additionally, the sampling sites with high POC/TSS but low $C_{ALG}$ values suggest high amounts of terrestrial carbon (as litter or soil organic matter) arriving from runoff to the rivers. On the other hand, the POC dilution could be explained by (a) the addition of soil-mineral-derived suspended solids during the RS for runoff increases, and (b) the low POC/TSS values that are registered in the Usumacinta River. Similar findings have been cited in other tropical rivers (e.g., [15,18]). For example, some Asian rivers draining through highly eroded regions exhibited a POC/TSS ranging between 0.2 and 3.6%, reflecting, in the same way, the terrestrial sedimentary origin [13,57,61].

Our results show that the contrasting seasonal hydrology in the Usumacinta River strongly impacted the TSS and POC, as well as in its sources (allochthonous versus autochthonous). The inter-tropical convergence zone oscillation largely influences the climatic seasonality and rainfall pattern in the tropics [62]. However, despite the climate, the hydrologic seasonality in tropical rivers is not a rule [63]. The extreme seasonality in the Usumacinta basin in comparison with other tropical rivers may reflect additional forcing agents, such as tropical cyclones, which drive outstanding seasonal changes in the hydrologic conditions and sediment transport [36]. This intra-seasonal variability may also affect the POC supply to the river, but it needs to be explored in future research by means of more detailed sampling campaigns over the rainy season. Since tropical storm

events and interannual climatic oscillations, such as ENSO and NAO, disproportionately affect precipitation and sediment transport [31,34], their effects on POC fluxes should be addressed in the next few years.

### 4.2. Spatial Variation of POC Sources

As expected, the middle basin of the Usumacinta River received more allochthonous POC than the lower basin because it is dominated by high-relief forested landscapes. On average and according to global data, POC/PN ratios greater than 15 usually indicate terrestrial vegetation sources [64] or poorly degraded plant constituents in tropical soils (e.g., tree litterfall and foliage [65]). Hence, the highest POC/PN and lowest $C_{ALG}$/POC ratios recorded in the uplands of the basin, which correspond to the most forested section (M1 and M2), during the RS may imply that terrestrial vegetation debris constituted the principal POC source in the Usumacinta River. These findings differ to some extent from those found in other tropical rivers flowing in highly productive forests combined with monsoon climate and steep slopes. While other studies indicate that the bulk of POC comes mainly from the topsoil (matured organic matter, POC/PN ratio between 6 and 15, e.g., [15,66]), here it seems that the fresh vegetation litter would be the principal POC source in forested areas. This aspect is probably associated with the high carbon storage capacity of rainforest soils [67].

It is possible that with the increase in primary producers in the DS that the allochthonous POC in the middle Usumacinta basin mixed with autochthonous sources. In the middle basin, the observed POC/PN ratio decrease in the DS may reflect the mixing of vascular (terrestrial) and non-vascular (phytoplankton) plants [68]. Moreover, the increase in the algal proportion in POC could also support this finding.

The pristine reference site M2 did not show seasonal variation in the TSS and turbidity. The trapping effect of the well-preserved Lacandona rainforest ecosystem could be responsible for reducing rainfall erosivity during storm events. The low concentrations of suspended solids in rivers that were least affected by farming suggest that the TSS had a significant plant litter fraction arriving from the soil surface [69]. Alternatively, since the POC/TSS ratio was high in both seasons, the falling litter of overhanging riparian vegetation may continuously in M2.

The increase in the algal proportion within the POC in the lower basin sites denotes a significant autochthonous contribution in both seasons. The POC/PN ratio also indicates an autochthonous contribution in the lower basin for both seasons since the ratios (<10) match the values reported for algae in diverse aquatic environments [65,69–71]. The low turbidity and water discharge, high residence times, and higher water temperatures favor algal development in the lower Usumacinta basin sampling sites. Other tropical rivers have higher algal growth under these environmental conditions during the drier stages [72]. Primary production in floodplains or lentic-like environments also enhances autochthonous organic matter inputs to the rivers [73]; for instance, the lower-basin floodplains of the Orinoco River display nutrient-rich waters fostering higher algal contributions [74]. There was an increase in Chl-a concentration in the dry season in floodplain wetlands and lakes connected to the Usumacinta River in the lower basin because of the high nutrient availability and large residence time [41]. Commonly, the autochthonous POC produced in floodplains during dry seasons reaches the rivers during wet seasons [12,75]. Thus, the high algal contribution observed in both seasons in this study could alternate between the river production and floodplain contribution.

In summary, the fluvial POC in the Usumacinta River was mostly allochthonous. The particulate organic matter arriving at the river differs seasonally in the forested section of the basin from a mixture composed of terrestrial vegetation and algal particles in the DS to one composed of fresh terrestrial vegetation with soil organic matter in the RS. The seasonality of the organic matter sources likely has implications for the Usumacinta River metabolism. Organic matter derived from soils is usually enriched in refractory C [76,77], while algae have a large proportion of labile substances, such as proteins and

carbohydrates [78]. Thus, in the Usumacinta River, carbon metabolism is fueled in the DS by inputs of fresh labile organic matter, which is more common in high water temperatures; by contrast, the more recalcitrant nature of organic matter reaching the river during the RS hinders the detrital pathway.

*4.3. POC Concentration and Yield Variabilities in the Usumacinta River Basin*

The POC concentration and flux increase observed in the middle basin during the RS represent the allochthonous sources delivered through runoff. The C litterfall from tropical forests transported toward the rivers constitutes a significant portion of the total fluvial POC transport, particularly in the RS in forested headwaters compared with plateau streams [79]. Tropical forested areas are major contributors to organic carbon flux with regard to other landscape units [67]. Litterfall from the riparian vegetation could also be exceptionally high [10].

Downstream of the tributary M1, the POC concentration decreased in both seasons because of a dilution effect caused by the tributaries. Some tributaries at the middle basin exhibited higher TSS and Q loads but a low C content. Although the flow of the Chixoy River is restricted by a hydroelectric dam built in the Guatemalan highlands, which should retain a considerable amount of fluvial sediments from the upper basin, the sediment flux toward the Usumacinta River is still relatively high [80]. Other factors, such as land-use changes (e.g., deforestation and agriculture), strongly decrease the soil organic matter and increase soil erosion and sediment transport to the La Pasion and Chixoy rivers. The loss of forest cover in the Usumacinta middle basin is more intense than in the upper and lower basins [38].

*4.4. POC Export to the Gulf of Mexico*

According to the mass balance and different to what was expected, the POC flux in the lower basin depended on primary producers and the hydrology dynamics. Therefore, the lower basin during the DS acted as a POC source due to the high autochthonous contribution, while in the rainy season it acted as a POC sink. The flooding occurring during the RS overflowed the main channel of the Usumacinta River, and a large fraction of the POC settled in the adjacent floodplains. During the RS, the flow connectivity with adjacent lakes and wetlands increases strongly in the lower Usumacinta River [22], as is commonly observed in other seasonal systems [81,82]. Indeed, the POC retention in the lower basin accounts for 23% of the middle basin delivery.

POC inputs enhance the mineralization in floodplains and the algae increase primary production through the use of recycled organic matter [41]; subsequently, the deposited allochthonous POC becomes new autochthonous matter. This fresh organic matter is possibly incorporated into the river during the DS when the water level decreases and the flow returns to the main channel. For instance, the lower basin retained 142.8 t of the allochthonous POC derived from the middle basin. Subsequently, 15.5 t of the POC was incorporated into the river again as autochtho-nous matter in the DS. Other tributaries (e.g., San Pedro River) draining through ex-tensive wetlands in the lower basin, supply autochthonous POC that adds to the final export of the Usumacinta River to the Gulf of Mexico.

Our results reveal that phytoplankton growth represents between 9 and 18% of the POC produced in the lower basin. This fraction concurs with the addition of organic matter by aquatic producers in the lower Amazon River (17% [83]), reaching 12% in the floodplain wetlands connected to the main river [18].

The POC export from the tropical Usumacinta River to the Gulf of Mexico is highly seasonal. The POC export increased by almost nine times in the RS with regard to the DS, which is larger than that reported for other temperate, larger rivers, such as the Mississippi River in the United States (1.6 to 5 times [54]; Table 4), but closer to that of other tropical monsoonal rivers, such as the Western Ghats in India (12 times [60]). However, the POC export from the Usumacinta River was 35 times lower than that of the Amazon river

export [18]. These values are consistent due to the fact that the Amazon River has the highest water discharge (6300 km$^3$ yr$^{-1}$) and drainage area (6300 × 10$^{-3}$ km$^2$) [84]. The POC export in the Usumacinta River (Q = 147 km$^3$ yr$^{-1}$) is similar to that of the Maroni River at the border between French Guiana and Suriname (Table 4), which has an annual Q of 63.7 km$^3$ yr$^{-1}$, a smaller drainage area (0.07 × 10$^{11}$ vs. 0.11 × 10$^{11}$ km$^2$), and high forest cover [85]. Additionally, being smaller in terms of water discharge (~5 times) and drainage area (~30 times) than the Mississippi River, the Usumacinta River showed a noteworthy proportional POC delivery to the Gulf of Mexico. For instance, although the POC export calculated in this study was lower than the Mississippi River POC transport (~1013 t C d$^{-1}$ in the DS and ~5287 t C d$^{-1}$ in the RS; [54]), it represented up to ~10% of the Mississippi River daily POC export in the DS and ~20% in the RS.

The POC concentration observed in the Usumacinta River is similar to or higher than that of other tropical rivers, such as the Maroni River [85] and Oyapock River, Senegal, Africa [12]. The Usumacinta River's POC concentration nearly doubled the range of the Orinoco River, Venezuela, and Colombia, despite the water discharge differences (Table 4). Differently, the Usumacinta River POC concentration range is lower than that of the whitewater Amazon River tributaries [18]; the Mississippi River [54]; and some highly turbid tropical rivers with high TSS loads (>100 mg L$^{-1}$), such as the Tana [86] and Yellow rivers [87].

**Table 4.** Water discharge (Q), POC concentrations, and flux of some rivers in the world.

| River | Country | Zone | POC (mg L$^{-1}$) | POC (t C d$^{-1}$) | Q (m$^3$ s$^{-1}$) | Ref. |
|---|---|---|---|---|---|---|
| Usumacinta | Mexico | Trop | 0.48–4.7 | 109–926 | 23–5924 | This study |
| Maroni | Guyana | Trop | 0.45–4.56 | 17–1266 | 178–4634 | a |
| Oyapock | Guyana | Trop | 0.64–3.16 | 9–279 | 141–2564 | a |
| Orinoco | Venezuela | Trop | 0.17–2.29 | 200–7500 | 500–70,000 | b |
| Amazon | Brazil | Trop | 0.27–26.8 | 4060–33,523 | 28,000–209,000 | c |
| Senegal | Africa | Trop | 0.2–4 | | ≤1700 | d |
| Tana | Kenya | Trop | 0.23–119.8 | | 123–208 | e |
| Mississippi | USA | Temp | 1.9–9.7 | 1013–5287 | 13,196–29,733 | f |
| Yellow | China | Temp | 4.6–92.4 | | | g |

Trop: tropical; Temp: temperate. a [85]; b [88]; c [18]; d [12]; e [53]; f [54]; g [87]. POC and Q ranges were obtained from major tributaries and the main channel; POC flux at location closest to the river mouth.

The northern Gulf of Mexico receives an important amount of particulate matter and dissolved organic carbon, nutrients, and freshwater from the Mississippi River [89], leading to the development of hypoxic marine zones [90]. Hypoxia is an environmental phenomenon where the concentration of dissolved oxygen in the water column decreases to a level that can no longer support living aquatic organisms. Hypoxic areas, or "dead zones," are likely to develop in the southern Gulf of Mexico due to the POC, nutrient, and freshwater delivery of the Usumacinta and Grijalva Rivers. The Usumacinta River mouths are highly eutrophicated (mesotrophic to eutrophic) in the upper water column, which consumes most oxygen derived from phytoplankton, causing lower oxygen saturations in coastal waters [91]. The information on the hypoxia and potential dead zone formation in the Usumacinta River influence zone should not be discarded and more investigatory studies need to be carried out on this topic [92].

## 5. Conclusions

The analysis of the seasonal dynamics of fluvial POC sources, concentration, and flux in the Usumacinta River basin relies on spatial differences and mechanisms that were determined using hydrogeomorphic features at the basin scale. The middle basin (steep slopes, forested) displayed a higher seasonal effect on POC concentration and source due to the potential erosion of terrestrial inputs compared to the lower basin (plateau, wetlands,

and floodplains). On the other hand, the algal-derived autochthonous POC seemed to lead to a steady concentration throughout seasons in the lower basin. The POC flux difference between seasons, which was high compared with that of other large river basins, was in agreement with the hydrological contrast in the basin. The "Pantanos de Centla" wetlands located in the lowlands displayed a crucial alternate role in the Usumacinta River's POC discharge into the Gulf of Mexico. The wetlands shifted between a sink zone of allochthonous POC in the RS (almost one-fifth of the POC transported downstream) and a source of algal fresh POC in the DS (nearly 70% of the final POC discharge). This seasonal effect caused around 20% of the total POC export to the ocean. The conservation of the large floodplains and wetlands covering most of the lower basin of the Usumacinta River is crucial for regulating the flow of POC to the southern Gulf of Mexico and maintaining a supply of labile organic matter to sustain marine productivity.

**Author Contributions:** Conceptualization, D.C.-L., J.A., D.C.-G., I.F.S.-R. and L.A.O.; Data curation, D.C.-L., D.C.-G. and I.F.S.-R.; Formal analysis, D.C.-L., J.A., D.C.-G., I.F.S.-R., F.G.-O. and S.S.-C.; Funding acquisition, J.A.; Investigation, D.C.-L., J.A., D.C.-G., I.F.S.-R. and L.A.O.; Methodology, D.C.-L., J.A., D.C.-G., I.F.S.-R. and L.A.O.; Project administration, J.A.; Resources, J.A.; Software, D.C.-L., D.C.-G. and I.F.S.-R.; Supervision, J.A.; Validation, D.C.-L., J.A., D.C.-G., I.F.S.-R., F.G.-O., S.S.-C. and L.A.O.; Writing—original draft, D.C.-L., J.A., D.C.-G., I.F.S.-R. and L.A.O.; Writing—review and editing, D.C.-L., J.A., D.C.-G., I.F.S.-R., F.G.-O., S.S.-C. and L.A.O. All authors have read and agreed to the published version of the manuscript.

**Funding:** This research was financially supported by the FORDECYT-CONACYT project 273646 "Fortalecimiento de las capacidades científicas y tecnológicas para la gestión territorial sustentable de la Cuenca del Río Usumacinta y su Zona Marina de Influencia (CRUZMI), así como su adaptación ante el cambio climático" and UNAM-PAPIIT project IN216818 "Flujos de carbono, nutrientes y sedimentos en un sistema lótico tropical." This research was also financially supported by the Programa de Investigación en Cambio Climático (PINCC 2020-2021) through project "Cuerpos acuáticos epicontinentales: papel en la dinámica del carbono y emisiones de gases de efecto invernadero en México."

**Institutional Review Board Statement:** Not applicable.

**Informed Consent Statement:** Not applicable.

**Data Availability Statement:** The data that support the findings of this study are available from the corresponding author (J.A.) upon reasonable request.

**Acknowledgments:** We would like to thank the "Posgrado en Ciencias del Mar y Limnología," UNAM, and CONACYT grants awarded to I.F.S.-R., D.C.-G. and J.D.C.-L. Jorge Ramírez and Julio Diaz helped with fieldwork and data collecting and processing. Natura y Ecosistemas Mexicanos AC foundation supported with logistic services at the Chajul Biological Station.

**Conflicts of Interest:** The authors declare no conflict of interest.

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
