# Peer review of "Particulate Organic Carbon in the Tropical Usumacinta River, Southeast Mexico: Concentration, Flux, and Sources"

_water, doi:10.3390/w13111561_

Round 1

Reviewer 1 Report

Very interesting work. Topics of work, results, discussion and conclusions at a good level. Suitable for publication in its current form.

Author Response

The manuscript was updated according to the referees' comments and edited for the English language.

Reviewer 2 Report

Dear Authors,

It is my opinion that your manuscript is suitable for pubblication in the journal, the paper reflects the requirements of "Water". The introduction section provide a good background for the readera. Moreover, the section Material and Methods was very accurate and precise showing all the sampling methods in detail. Both results and discussion were clearly presented. 

good luck for your submission  

Author Response

(The authors gave the same response as above.)

Reviewer 3 Report

Review of Cuevas-Lara et al study of particulate organic carbon in tropical Usumacinta River …

General comments:

The quality of the English in the manuscript is poor and at times unreadable. This causes problem in determining whether the science is acceptable, or no. f had been sent aversion as word document I would have tried to correct at least some of the English but since Water sends out a pdf, it is not possible to make such changes. Before the manuscript can be properly reviewed it needs to be read by a native English speaker so than it can be properly understood by the reviewers (and eventually if accepted) by the readers. For now, I have simply marked the line numbers in which there is a problem with the English in my detailed comments below

Specific comments:

The first two paragraphs make unreasonable and extravagant claims of the importance of this work to the global C cycle. It is a study worth doing but I do not accept that for instance rivers play a crucial role in global biogeochemical C cycles or POC arriving through rivers regulates the atmospheric C reservoir!

Actually, the rainy season sampling was carried out in the beginning of the dry season. I have no idea if that is a problem i.e., both samplings were during the dry season.

Problems with English in lines 33,34, 36,38. 51, 54, 56, 59, 62, 66, 71, 75, 77, 79,84, 89, 110, 117, 119, 125, 137, 138, 139, 146 and 154

Line 185 mgC not Mg

Equations between lines 191 and 196 are all missing key descriptors. Thus CALG/POC does not equal CALG/POC x  100% .

Results section:

This section lists a very large number of detailed statistical relationships. Since I read/review from the beginning to the end, I am left asking whether all of these values are going to be used in order to develop the principal answers highlighted in the discussion.

Problems with English in lines 226, 228, 246 concentrations of what?, 250, 255, 256, 257, 258, oscillates implies a relationship that goes +-+-+-, 262, what is indirectly correlated?, line 279 not oxidant, 280, 282, 295, 336 should be kg km2, not capitals

Discussion

  A well written discussion consists of 2-4 clear answers to knowledge gaps defined in the introduction. In this text I am finding great difficulty seeing any clear ‘answers’ to any questions. This may of course be partly because the English is so badly written, or it might be because the data is not resulting in clear answers.

Specifically

Problems with English in lines: 361, 364,

367 I am sure the yellow is not subtropical; I think the Mississippi and the tana aren’t either.

369,370,371

373 where are the turbid flows?

Where are the plots of TSS vs POC that have a direct relationship?

374-381 why are you mentioning ideas and saying we will discuss later?

387,391, 402,404,407

413 they may affect POS supply to the river but do they?

441, 446 what is lentic,453 alternate between?

455, 456, 457, 458, 460

470

Where did litterfall represents a substantial part of fluvial CO2 emission come from? Is it relevant here? You have no estimate of CO2 flux?

Line 490, 502, line 510 and 511 what are Mg units wrong.

Line 524 what is efflux

Line 530 don’t use significant unless it is a statistical test.

Line 536 yes freshwater is less dense than seawater!

Section on hypoxia: This is driven by nutrients which cause local PP which then respires and consumes oxygen. This study has no nutrient data

Author Response

REFEREE COMENT

ANSWER

The quality of the English in the manuscript is poor and at times unreadable. This causes problem in determining whether the science is acceptable, or no. f had been sent aversion as word document I would have tried to correct at least some of the English but since Water sends out a pdf, it is not possible to make such changes. Before the manuscript can be properly reviewed it needs to be read by a native English speaker so than it can be properly understood by the reviewers (and eventually if accepted) by the readers. For now, I have simply marked the line numbers in which there is a problem with the English in my detailed comments below

The updated version of the ms was English language edited through MDPI Author Services (certification attached) prior re-submission

INTRODUCTION SECTION

The first two paragraphs make unreasonable and extravagant claims of the importance of this work to the global C cycle. It is a study worth doing but I do not accept that for instance rivers play a crucial role in global biogeochemical C cycles or POC arriving through rivers regulates the atmospheric C reservoir!

We re-wrote -moderate- the indicated paragraphs to avoid misinterpretations.

MATERIALS AND METHODS SECTION

Actually, the rainy season sampling was carried out in the beginning of the dry season. I have no idea if that is a problem i.e., both samplings were during the dry season.

Sampling was carried out at “extreme” conditions, this is, 1) just at the end of the dry season when the water level and river flow were minimal (May), and 2) at the end of the rainy season when the water level and river flow were close to maximum (November). In October, the river is at maximum flow, however, it was not possible to sample under such harsh/dangerous conditions (e.g., is not possible to go upstream against the current in motorboats nor even keep a fixed position to take vertical samples; the river transport large pieces of three trunks). If the referee prefers, the rainy and dry season could be replaced by 1) lower water level and flow and 2) higher water level and flow.

Line 185 mgC not Mg

Mg means “megagram”, this is, 1,000,000 grams or a metric ton (1,000 kg). It is a unit in the SI that water journal recommends. Nonetheless, to avoid confusion we changed to tons (t or T).

Equations between lines 191 and 196 are all missing key descriptors. Thus CALG/POC does not equal CALG/POC x 100% .

Key descriptors were added as requested.

RESULTS SECTION

This section lists a very large number of detailed statistical relationships. Since I read/review from the beginning to the end, I am left asking whether all of these values are going to be used in order to develop the principal answers highlighted in the discussion.

We largely modified the ms following the referees’ suggestions and comments. The relationships that provide little information to the discussion were removed.

concentrations of what?

We clarified it in the sentence of the text as requested.

250, 255, 256, 257, 258, oscillates implies a relationship that goes +-+-+-

We changed the verb in the sentence of the text as requested.

262, what is indirectly correlated?

The paragraph was rewritten for clarity.

line 279 not oxidant,

We changed oxidant to “oxidizing environment”

should be kg km2, not capitals

We changed the capital letter in the units as requested.

DISCUSSION SECTION

A well written discussion consists of 2-4 clear answers to knowledge gaps defined in the introduction. In this text I am finding great difficulty seeing any clear ‘answers’ to any questions. This may of course be partly because the English is so badly written, or it might be because the data is not resulting in clear answers.

The ms was largely modified for clarity and English was largely improved. We are sure the referee will be now able to clearly follow the discussion.

367 I am sure the yellow is not subtropical; I think the Mississippi and the tana aren’t either.

The referee is right. We re-wrote the paragraph.

373 where are the turbid flows?

In the main channel. We included this information in the text.

Where are the plots of TSS vs POC that have a direct relationship?

The plot was added as requested (Fig. 3).

374-381 why are you mentioning ideas and saying we will discuss later?

We removed this part to avoid confusion.

413 they may affect POS supply to the river but do they?

Yes, tropical cyclones increase POC supply as evidenced in the rainy season when tropical storms take place. It is now indicated in the text.

441, 446 what is lentic?

In Limnology it is well-known both terms: lentic and lotic. Inland water bodies are classified into two categories: lotic (“flowing water”, rivers, streams) and lentic (“still or standing water”, lakes, ponds). Nonetheless, rivers display zones with different water velocity distinguishing, for example, “pools”, zones more likely standing/lentic waters, and “rapids”, zones more likely fast flowing/lotic waters.

453 alternate between?

As mentioned in the text, alternate between algal PP taking place in the river and algal PP taking place in floodplains.

Where did litterfall represents a substantial part of fluvial CO2 emission come from? Is it relevant here? You have no estimate of CO2 flux?

The referee is right, we did not measure CO2 emission. The sentence was removed.

Line 490, 502, line 510 and 511 what are Mg units wrong.

Mg means “megagram”, this is, 1,000,000 grams or a metric ton (1,000 kg). It is a unit in the SI that water journal recommends. Nonetheless, to avoid confusion we changed to tons (t or T).

Line 524 what is efflux

We used the term efflux as the flowing of the particles out of the system (in this case out of the river into the Gulf of Mexico). Nonetheless, we have changed it to “export” to avoid confusion.

Line 530 don’t use significant unless it is a statistical test.

We changed the word in the sentences to avoid confusion.

Line 536 yes freshwater is less dense than seawater!

We deleted the clarification in the sentence in agreement with the comment.

Section on hypoxia: This is driven by nutrients which cause local PP which then respires and consumes oxygen. This study has no nutrient data

The referee is correct, we present no data on nutrients. The comment regarding hypoxia is based on the large POC concentrations we measured, and references therein mentioned. The relevance and implications of the well-studied hypoxic zones in the northern Gulf of Mexico, make it interesting to mention. The Grijalva-Usumacinta is the most important river system in Mexico. The Grijalva (second largest river) polluted waters join the Usumacinta (first largest river) before jointly discharging in the southern Gulf of Mexico. Along with nutrients that increase PP (as the referee mentioned), POC supply by the rivers is also respired which adds to DO consumption and potentially generates hypoxic zones commonly found associated with the large river mouths. We re-wrote this paragraph.